# AMX0035 Mitigates Oligodendrocyte Apoptosis and Ameliorates Demyelination in MCAO Rats by Inhibiting Endoplasmic Reticulum Stress and Mitochondrial Dysfunction

**DOI:** 10.3390/ijms26083865

**Published:** 2025-04-19

**Authors:** Li Zhang, Cunhao Bian, Yusen Wang, Ling Wei, Shanquan Sun, Qian Liu

**Affiliations:** 1School of Basic Medical Sciences, Chongqing University of Chinese Medicine, Chongqing 402760, China; 15111938438@163.com; 2Department of Forensic Medicine, Chongqing Medical University, Chongqing 400016, China; forensicbch@163.com (C.B.); wangyun111001017@163.com (Y.W.); 3Institute of Neuroscience, Chongqing Medical University, Chongqing 400016, China; 13996119601@163.com

**Keywords:** post-stroke cognitive impairment, demyelination, oligodendrocytes, endoplasmic reticulum stress, mitochondrial dysfunction

## Abstract

Post-stroke cognitive impairment (PSCI) is a common complication of strokes and is associated with the demyelination of nerve fibers. AMX0035, a drug currently used to treat motor neuron diseases, may aid in preventing oligodendrocyte apoptosis and alleviating demyelination by targeting the pathways involved in ERS and mitochondrial dysfunction. All animals were randomly divided into four groups: the sham, sham+AMX0035, middle cerebral artery occlusion (MCAO), and MCAO+AMX0035 group. The Morris water maze was used to test cognitive function, and changes in myelin structure in the brain were investigated using transmission electron microscopy (TEM), Luxol fast blue (LFB) staining, and myelin basic protein (MBP) immunofluorescence staining. Western blot was performed to detect proteins associated with ER stress and mitochondrial dysfunction, and double-labeling immunofluorescence was utilized to localize oligodendrocytes and apoptosis-related proteins. Neurological function scores and TTC staining confirmed the successful establishment of the MCAO rat model. The Morris water maze experiment revealed impaired cognitive function in MCAO rats, which significantly improved following the AMX0035 intervention. TEM and LFB staining showed the disrupted myelin structure in the MCAO group, while AMX0035 effectively ameliorated this myelin damage. Immunofluorescence examination and Western blot revealed the decreased expression of MBP in MCAO rats, increasing with AMX0035 treatment. TUNEL staining demonstrated increased cell apoptosis in MCAO rats, which was reduced following AMX0035 therapy. Western blot detected significant increases in proteins associated with the ER stress pathway and proteins linked to mitochondrial dysfunction in the MCAO group, all of which were downregulated after AMX0035 intervention. Double-labeling immunofluorescence staining revealed a significant increase in the number of cytochrome c^+^ and caspase 12^+^ oligodendrocyte cells in MCAO rats, which decreased after AMX0035 administration. The activation of ER stress and mitochondrial dysfunction pathways following MCAO led to oligodendrocyte damage and apoptosis. AMX0035 can inhibit these pathways, reduce oligodendrocyte apoptosis, and alleviate demyelination, thereby improving PSCI.

## 1. Introduction

Stroke, referring to a condition in which brain tissue is damaged due to the abrupt rupture or blockage of cerebral blood vessels, stands as the second most common cause of death and the third leading cause of disability worldwide [1]. In China, it is the leading cause of death [2,3]. The two main types of stroke are ischemic and hemorrhagic, with ischemic stroke representing over 80% of cases [4,5].

A common complication of stroke is post-stroke cognitive impairment (PSCI), which results from complex pathophysiological mechanisms. White matter injury has been identified as a major factor contributing to PSCI, primarily characterized by the demyelination of central nervous system (CNS) nerve fibers and axonal damage [6,7,8]. Demyelination severely impacts neurological function, including motor and cognitive abilities, and represents a major pathological change in PSCI. Therefore, protecting white matter integrity and promoting axonal myelin regeneration are considered crucial strategies for treating PSCI.

Oligodendrocytes play a crucial role in the development of myelin within the CNS [9]. Studies have shown that stroke can lead to oligodendrocyte apoptosis, thereby triggering demyelination [10,11]. Both endoplasmic reticulum (ER) stress and mitochondrial dysfunction significantly contribute to this apoptotic process. ER stress triggers apoptosis by activating signaling pathways, including the expression of caspase 12, which promotes cell death [12,13]. Similarly, mitochondrial dysfunction facilitates apoptosis through the release of cytochrome c [14,15]. During ER stress, Ca^2+^ ions are transferred from the ER to the mitochondria. The resulting Ca^2+^ overload causes mitochondrial membrane depolarization, leading to the release of cytochrome c, which, in turn, activates apoptotic pathways and ultimately induces cell death. Previous research from our group has demonstrated that the interaction between ER stress and mitochondrial dysfunction induces oligodendrocyte apoptosis and contributes to demyelination following spinal cord injury [9,16]. Therefore, ER stress and mitochondrial dysfunction are key factors in initiating oligodendrocyte apoptosis and promoting demyelination following a stroke. These mechanisms contribute significantly to white matter damage, impairing myelin integrity and affecting neural connectivity and function.

AMX0035 is a drug composed of two components: sodium phenylbutyrate (PBA) and taurursodiol (TUDCA). PBA effectively inhibits ER stress and mitigates neuronal damage [17,18], while TUDCA exerts neuroprotective effects by inhibiting ER stress and modulating mitochondrial function [19,20]. Existing studies have confirmed that AMX0035 can target multiple signaling pathways, including the inhibition of ER stress and mitochondrial dysfunction, demonstrating significant neuroprotective effects [21]. However, the effects of AMX0035 on improving PSCI and its underlying mechanisms require further exploration.

This study aims to establish a middle cerebral artery occlusion (MCAO) rat model of cerebral ischemia and administer AMX0035 as an intervention. Using behavioral and molecular biology approaches, the objectives are as follows: (1) to determine whether AMX0035 improves post-stroke cognitive impairment (PSCI); (2) assess whether AMX0035 reduces oligodendrocyte apoptosis by inhibiting ER stress and mitochondrial dysfunction pathways, thereby alleviating central nervous fiber demyelination; and (3) clarify the underlying mechanisms of action of AMX0035.

## 2. Results

### 2.1. Successful Establishment of the MCAO Rat Model

In the MCAO group, approximately three rats (~10%) died due to surgical complications. Neurological function scores were evaluated in the MCAO group, and rats with scores of one or four, as well as those that did not survive the surgery, were excluded from further analysis. Following the neurological assessment, an additional three rats were excluded from the MCAO group due to failure to meet the neurological criteria. Furthermore, three rats from the MCAO group were randomly selected for TTC staining. As shown in Figure 1, significant infarct areas were observed in the brain tissue of the MCAO group, confirming the successful establishment of the model.

### 2.2. AMX0035 Improves Cognitive Function in MCAO Rats

Cognitive function was evaluated using the Morris water maze. The movement trajectories of rats in each group are shown in Figure 2a. Prior to training, there were no significant differences in swimming speed among the groups (Figure 2b). The escape latency during days 1 to 5 of training was significantly longer in the MCAO group compared to the sham and sham+AMX0035 groups (Figure 2c). Additionally, the time spent on the platform was significantly reduced in the MCAO group on day 5 (Figure 2d), indicating cognitive decline. After the AMX0035 intervention, the escape latency was significantly reduced (Figure 2c), and the time spent on the platform was significantly increased (Figure 2d). The results from the water maze experiment demonstrate that AMX0035 improved cognitive function in the MCAO group. Specifically, one-way ANOVA showed a significant difference among the groups (F = 6.208, *p* = 0.0021, R^2^ = 0.3830).

### 2.3. AMX0035 Alleviates Demyelination in the Brain Tissue of MCAO Rats

Following AMX0035 treatment, demyelination was significantly alleviated. LFB staining results (Figure 3a) showed a disorganized myelin morphology in the corpus callosum of the MCAO group, with lighter staining and vacuolation, reflecting marked demyelination. After AMX0035 treatment, demyelination was mitigated.

TEM observations of the corpus callosum and hippocampus revealed structural changes in myelin (Figure 3b). The sham and sham+AMX0035 groups showed well-organized and structurally intact myelin, whereas the MCAO group exhibited thinning and disorganized myelin, indicative of demyelination (as marked by black arrows).

The TEM observations of the corpus callosum and hippocampus revealed structural changes in myelin (Figure 3b). The sham and sham+AMX0035 groups showed well-organized and structurally intact myelin, whereas the MCAO group exhibited thinning and disorganized myelin, indicative of demyelination (as marked by black arrows). Quantitative analysis of the myelin g-ratio in the corpus callosum and hippocampus further supports these observations. The g-ratio was significantly increased in the MCAO group compared to the sham group, indicating reduced myelin thickness relative to the axon diameter (Figure 3c,d). Treatment with AMX0035 partially restored the g-ratio toward normal levels, suggesting a protective effect against demyelination.

Immunofluorescence (IF) analysis (Figure 4) revealed a significant reduction in myelin basic protein (MBP) density in the cerebral cortex, corpus callosum, and hippocampus of the MCAO group compared to both the sham and sham+AMX0035 groups, while AMX0035 therapy resulted in increased MBP density in these regions.

### 2.4. AMX0035 Inhibits Cell Apoptosis in the Brain Tissue of MCAO Rats

TUNEL staining results (Figure 5) revealed a significant increase in apoptotic cells in the brain tissue of the MCAO group compared to the sham and sham+AMX0035 groups. AMX0035 treatment markedly reduced the number of TUNEL-positive cells, indicating its potential anti-apoptotic effect in ischemic brain tissue. These findings suggest that AMX0035 can attenuate neuronal apoptosis following MCAO. To further investigate whether AMX0035 specifically inhibits the apoptosis of oligodendrocytes and to explore its underlying molecular mechanisms, immunofluorescence double labeling will be performed in subsequent experiments.

### 2.5. AMX0035 Reduces Endoplasmic Reticulum Stress in the Oligodendrocyte of MCAO Rats

ER stress plays a critical role in ischemia–reperfusion injury in MCAO brain tissue. Western blot results from this study showed the significant upregulation of ER stress pathway proteins, including p-PERK, p-IRE1, GRP78, CHOP, cytochrome c, and caspase 12 in the corpus callosum of the MCAO group compared to the sham and sham+AMX0035 groups. The AMX0035 intervention significantly inhibited the expression of these ER stress pathway proteins (Figure 6a–c).

To investigate whether ER stress occurs in oligodendrocytes following MCAO, contributing to cell damage and apoptosis, this study conducted double-label immunofluorescence staining using the ER stress-related apoptotic marker caspase 12 and the oligodendrocyte marker CNPase. As shown in Figure 7, only a few caspase 12^+^ oligodendrocytes were sporadically distributed in the brain tissue of the sham and sham+AMX0035 groups, whereas the number of caspase 12^+^ oligodendrocytes was significantly elevated in the brain tissue of MCAO rats. Following AMX0035 treatment, the number of caspase 12^+^ oligodendrocytes in MCAO rat brain tissue was significantly reduced, demonstrating that AMX0035 can inhibit ER stress and associated apoptosis in oligodendrocytes within MCAO rat brain tissue.

### 2.6. AMX0035 Inhibits Mitochondrial Dysfunction in OLs of MCAO Rats

Mitochondrial dysfunction is a crucial pathological process in ischemic brain injury. Cytochrome C is a protein located in the mitochondrial inner membrane, involved in oxygen utilization and ATP synthesis in the electron transport chain. During brain ischemia, mitochondrial membrane integrity is compromised, leading to the release of cytochrome c into the cytoplasm, activating the caspase cascade and resulting in cell death. As shown in Figure 6a,c, compared to the sham and sham+AMX0035 groups, the expression of mitochondrial dysfunction-related apoptotic protein cytochrome c was significantly upregulated in the corpus callosum and hippocampus of the MCAO group, which was significantly downregulated following AMX0035 treatment.

To assess whether mitochondrial dysfunction occurs in oligodendrocytes following MCAO, this study performed double-label immunofluorescence staining using cytochrome c and the oligodendrocyte marker CNPase. The results (Figure 8) showed that a small number of cytochrome c^+^ oligodendrocytes were scattered throughout the brain tissue in the sham and sham+AMX0035 groups, while the number of cytochrome c^+^ oligodendrocytes was significantly increased in the MCAO group. Following AMX0035 intervention, the number of cytochrome c^+^ oligodendrocytes in MCAO rat brain tissue was markedly reduced, indicating that AMX0035 effectively inhibits mitochondrial dysfunction in oligodendrocytes within MCAO rat brain tissue.

### 2.7. AMX0035 Inhibits the Interaction Between ER Stress and Mitochondrial Dysfunction in the Brain Tissue of MCAO Rats

The interaction between ER stress and mitochondrial dysfunction can mediate apoptotic cell death. To explore the interplay between ER stress and mitochondrial dysfunction in regulating oligodendrocyte apoptosis following MCAO, dual-label immunofluorescence was used to detect the co-localization of caspase 12 and cytochrome c (Figure 9). The results indicate that the number of co-localized cells expressing caspase 12 and cytochrome c was significantly increased in the brain tissue of the MCAO group compared to the sham and sham+AMX0035 groups. AMX0035 administration significantly reduced the number of these co-localized cells, demonstrating that AMX0035 effectively inhibits the interaction between ER stress and mitochondrial dysfunction in the brain tissue of MCAO rats.

## 3. Discussion

PSCI is a prevalent complication that profoundly impacts patients’ quality of life and prognosis. Understanding PSCI’s underlying mechanisms is essential for early identification and intervention to enhance patient outcomes. Research indicates significant white matter abnormalities in PSCI patients and animal models, primarily characterized by demyelination and axonal injury. Cognitive function relies on intact myelin structures, making demyelination detrimental to neural conduction and cognitive abilities [22]. MBP, synthesized by oligodendrocytes, plays a vital role in myelination by promoting the formation of major dense lines, and its degradation is a hallmark of demyelination [23]. This study explores the therapeutic effects of AMX0035 on cognitive impairment in an MCAO rat model, demonstrating its potential to improve neurological function through various mechanisms. Behavioral tests confirmed cognitive deficits correlated with myelin damage, supported by TEM, LFB staining, and MBP analyses, which revealed significant demyelination affecting neural conduction.

After a stroke, secondary neurological injury involves various complex pathophysiological processes, such as ER stress, mitochondrial dysfunction, oxidative stress, and apoptosis [24,25,26]. The ER is crucial for protein modification, lipid synthesis, and calcium regulation [27,28,29,30]. When ER homeostasis is disrupted, it results in biochemical changes, leading to apoptotic pathways that worsen brain injury [31]. The mitochondria are essential for energy supply and apoptosis; ischemic conditions cause mitochondrial damage, enhancing reactive oxygen species (ROS) production and cytochrome c release, thereby promoting brain damage [29,32]. This study demonstrated the significant upregulation of ER stress and mitochondrial dysfunction-related proteins in MCAO rat brain tissue, highlighting their roles in the development of PSCI.

Oligodendrocytes play a critical role in myelination within the CNS, making their health vital for effective neural conduction [33]. The damage or apoptosis of oligodendrocytes can result in demyelination or insufficient myelination, adversely affecting cognitive function. In the context of ischemic stroke, oligodendrocyte-induced demyelination contributes to white matter injury, underscoring the significance of oligodendrocyte health [29]. Research indicates that the apoptosis of oligodendrocytes following intracerebral hemorrhage (ICH) leads to the degradation of MBP and can trigger neuronal apoptosis [34,35]. Apoptosis can occur via several pathways, including the death receptor, mitochondrial, and ER pathways. Notably, the ER pathway is implicated in mediating oligodendrocyte apoptosis after ICH [36,37]. During ER stress, Ca^2+^ overload can disrupt mitochondrial membrane potential, resulting in cytochrome c release and the activation of apoptotic factors, leading to cell death [14,15]. This experiment utilized dual-label immunofluorescence staining to assess the ER stress-related apoptosis marker caspase 12, mitochondrial dysfunction-related protein cytochrome c, and the oligodendrocyte marker CNPase in the brain tissue of MCAO rats. These findings indicate an increase in caspase 12 and cytochrome c-positive oligodendrocytes, reinforcing the roles of ER stress and mitochondrial dysfunction in oligodendrocyte damage and apoptosis. Furthermore, the co-localization analysis of caspase 12 and cytochrome c revealed their interaction, highlighting their combined contributions to the pathological processes of PSCI, which lead to neuro-injury and cognitive impairment.

AMX0035, composed of PBA and TUDCA, has been shown to improve cognitive dysfunction and peripheral neuropathy caused by various cognitive diseases when used individually. PBA, as an ER stress inhibitor, effectively mitigates ER stress and reduces neuronal damage, which can alleviate oligodendrocyte apoptosis and improve demyelination [17,18,38,39,40]. It plays an essential role in ameliorating cognitive dysfunction caused by neurological injuries or diseases [41,42,43]. In cases of peripheral nerve injury, PBA can inhibit neuroinflammation, promoting axonal repair and myelination. In recent years, TUDCA has demonstrated therapeutic efficacy in traumatic brain injury, cerebral hemorrhage, and neurodegenerative diseases [44,45]. Studies demonstrate that TUDCA can decrease lipopolysaccharide-induced neuroinflammation, reduce apoptosis, and ameliorate synaptic dysfunction, therefore reducing cognitive [46,47]. TUDCA exerts neuroprotective effects by inhibiting ER stress and modulating mitochondrial function, which, in turn, suppresses cytochrome c release [19,20]. AMX0035 has been shown to simultaneously inhibit multiple signaling pathways related to ER stress, mitochondrial dysfunction, and oxidative stress, providing neuroprotection [21]. In our study, after AMX0035 intervention, the escape latency in the model group of rats significantly decreased, while the time spent on the platform markedly increased, indicating an improvement in cognitive function in the MCAO group. TEM, LFB staining, immunofluorescence examinations, and Western blotting for MBP demonstrated that AMX0035 effectively alleviated demyelination in these rats. Furthermore, AMX0035 intervention significantly inhibited apoptosis in the brain tissue of MCAO rats, reducing the expression of ER stress pathway proteins (p-PERK, p-IRE1, GRP78, CHOP, and caspase 12) and mitochondrial dysfunction pathway proteins (cytochrome c) in the corpus callosum and hippocampus, as well as the numbers of caspase 12^+^ and cytochrome c^+^ oligodendrocytes and co-localized cells expressing caspase 12 and cytochrome c in the brain. This study confirms that AMX0035 significantly improves cognitive function and alleviates demyelination in MCAO rats, with the primary mechanism linked to reducing oligodendrocyte damage and apoptosis by inhibiting ER stress and mitochondrial dysfunction pathways. These results suggest that AMX0035 could serve as a potential therapeutic agent for PSCI and related neurological disorders.

Our study acknowledges several limitations. Firstly, we concentrated primarily on oligodendrocyte apoptosis and demyelination after cerebral ischemia. Axons are also vital components of white matter, and future research should examine the impact of AMX0035 on neuronal apoptosis and axonal degeneration in the post-ischemic context. Secondly, further investigations should assess whether AMX0035 can enhance brain recovery through other pathways, including oxidative stress and anti-inflammatory mechanisms.

## 4. Materials and Methods

### 4.1. Experimental Animals

In total, 66 male-specific pathogen-free (SPF) Sprague Dawley (SD) rats, aged between 8 and 10 weeks and weighing 250–280 g, were purchased from the Experimental Animal Center of Chongqing Medical University. The rats were housed in a controlled environment at 22 °C with 60% humidity, in a 12 h light/dark cycle, and provided with ad libitum access to food and water. Following an acclimatization period, the experimental procedures were initiated. All experimental protocols were approved by the Ethics Committee of Chongqing University of Chinese Medicine (Approval No. [2023]30-01).

### 4.2. MCAO Model Establishment

The rats were randomly divided into the sham surgery group (n = 36) and the MCAO model group (n = 30). At the beginning of the experiment, a total of 66 rats were randomly assigned to the sham surgery group and the MCAO model group. To account for potential intraoperative mortality and model establishment failures in the MCAO group, we initially allocated 30 rats to this group. In the MCAO group, the blood flow of the middle cerebral artery (MCA) was interrupted by inserting a silicone thread. After anesthetizing the rats, the right common carotid artery (CCA), external carotid artery (ECA), and internal carotid artery (ICA) were exposed. The ECA was ligated and severed, and the silicone thread (0.36 mm, Beijing Xinong Technology, Beijing, China) was inserted through the CCA into the ICA until it reached the MCA origin to block blood flow. After 2 h of ischemia, the thread was withdrawn. The sham surgery group underwent the same procedures but without the insertion of the thread, with the remaining steps being identical. Post-surgery, the rats were provided with appropriate care, and their body temperature was maintained.

### 4.3. Neurological Function Assessment

Twenty-four hours after modeling, neurological function was assessed using a modified Longa scoring system. In the MCAO group, scores ranging from 1 to 4 indicated successful model establishment, with scores ranging from 2 to 3 representing rats with moderate MCAO that were deemed suitable for subsequent experiments.

### 4.4. TTC Staining

Twenty-four hours after thread withdrawal, three rats from each group were randomly chosen to collect brain tissue for TTC staining to assess the extent of cerebral infarction. The brains were cut into approximately 2 mm thick coronal sections and incubated with a 2% TTC solution for 30 min in a light-protected environment.

### 4.5. Screening of PSCI Rats and AMX0035 Intervention

Initially, the experimental animals were divided into a sham group and an MCAO group. After the successful establishment of the MCAO model, all experimental animals were randomly divided into the following four groups: (1) the sham surgery group (n = 13); (2) the sham surgery+AMX0035 group (n = 13); (3) the MCAO group (n = 13); and (4) the MCAO+AMX0035 group (n = 13). The sham surgery+AMX0035 group and the MCAO+AMX0035 group received daily intraperitoneal injections of AMX0035 (200 mg/kg) for three consecutive weeks. The sham surgery group and MCAO group received equivalent volumes of saline. Following this treatment period, cognitive function was assessed through behavioral testing. For the sham group and sham+AMX0035 group, 1 spare rat was included in the case of unforeseen complications; however, as there were no adverse events in the sham surgeries, the spare rat was not utilized in the final analysis.

### 4.6. Behavioral Experiments

In the Morris water maze experiment, rats’ learning and memory abilities are assessed to evaluate their cognitive function. Each rat undergoes four training trials per day, with the rat being gently placed in the water at one of four different starting points. The training intervals are set at 20 min, and the experiment is conducted over five consecutive days. The trial ends either when a rat successfully reaches the hidden platform and stays there for more than 2 s or when the rat fails to locate the platform within 1 min. Rats that do not find the platform within the allotted time are guided to it and allowed to remain on the platform for 15 s, ensuring that all rats successfully locate the platform during the experiment. Twenty-four hours after the final training trial, the platform is removed. Each rat is then placed in the water from a new starting position to assess its spatial memory. The rat must rely on spatial cues to locate the platform’s previous location rather than following a specific swimming path. At this stage, each rat is allowed to swim freely for 1 min. Video-tracking software (Smart 3.0 (RWD Life Science Co., Ltd., Shenzhen, China)) records the rat’s movement trajectory, escape latency, and time spent in the platform zone, which together provide a measure of spatial memory performance.

### 4.7. Transmission Electron Microscopy (TEM)

After completing the behavioral experiments, the rats were anesthetized with 2% sodium pentobarbital (1.5 mL/kg) via intraperitoneal injection. Immediately after thoracotomy, cold saline was administered to rapidly extract the brain tissue while minimizing animal distress. The extracted brain tissue was then preserved in a 2.5% glutaraldehyde fixative solution at −4 °C for subsequent TEM analysis. For TEM analysis, we quantified the g-ratio (the ratio of the inner axonal diameter to the total outer fiber diameter), which provides an objective measure of myelin thickness. A total of 20 axons per group were randomly selected and analyzed using ImageJ software v1.54q2.

### 4.8. Luxol Fast Blue (LFB) Staining 

A portion of the brain tissue was fixed in formalin for one week, sliced into paraffin sections, and subjected to LFB staining to observe the myelin structure in the brain tissues of each group.

### 4.9. Immunofluorescence Staining

Three rat brain tissues were fixed in 4% paraformaldehyde solution at 4 °C for 24 h, followed by dehydration in a sucrose gradient (10%→20%→30%). After dehydration, the tissues were embedded in the OCT compound and sectioned into 10 μm thick frozen coronal slices. The sections were air-dried and stored at −20 °C for later use. These sections were used for immunofluorescence staining. The primary antibodies included CNPase (rabbit, 1:100), cytochrome c (rabbit, 1:500), caspase 12 (rabbit, 1:100), and MBP (rabbit, 1:200). The secondary antibodies included FITC-conjugated goat anti-rabbit IgG (1:200) and Cy3-conjugated goat anti-rabbit IgG (1:300). For each brain, 3 coronal sections were analyzed. In each section, 10 randomly selected regions of interest (ROIs) were examined. For double-labeling analysis, the co-localization of markers was assessed using the co-localization analysis plugin in ImageJ, and the fluorescence intensity of double-positive cells was quantified. The number of double-labeled cells was manually counted in the selected ROIs.

### 4.10. Western Blot

The hippocampal and corpus callosum tissues of the rats were homogenized and centrifuged, and the total protein supernatant was collected. The protein concentration was measured using a BCA protein assay kit. Twenty micrograms of the total protein was separated via 6% and 12% SDS-PAGE gel electrophoresis and transferred to a PVDF membrane. The membrane was blocked at room temperature with 5% non-fat milk for 2 h and then incubated overnight with the following primary antibodies: CHOP (1:1000), MBP (1:4000), caspase 12 (1:1000), GRP78 (1:4000), cytochrome c (1:2000), tubulin (1:20,000), PERK (1:750), p-PERK (1:2000), IRE1 (1:2000), and p-IRE1 (1:1000). Secondary antibodies included goat anti-mouse IgG (1:10,000) and goat anti-rabbit IgG (1:10,000). Detection was performed using ECL chemiluminescence, and imaging was performed using a ChemiDoc imaging system v5.2.1. Protein levels were quantified using Image Lab 5.2.1 software.

### 4.11. Statistical Analysis

All experiments were conducted in triplicate. The data are presented as the mean ± standard deviation (SD). Statistical analyses were performed using GraphPad Prism software v8.0, with one-way analysis used for comparisons. Statistical significance was indicated as * *p* < 0.05, ** *p* < 0.01, *** *p* < 0.001, and **** *p* < 0.0001, with *p* < 0.05 considered statistically significant.

## 5. Conclusions

AMX0035, a novel therapeutic agent composed of PBA and TUDCA, has demonstrated neuroprotective potential in various neurological disorders. This study confirms that AMX0035 significantly enhances cognitive function and reduces demyelination in MCAO rats, primarily by inhibiting ER stress and mitochondrial dysfunction pathways, thereby reducing oligodendrocyte damage and apoptosis. These findings suggest that AMX0035 may be a promising drug for treating PSCI and related neurological disorders, particularly by modulating intracellular signaling pathways to protect neurons and preserve nervous system integrity. However, further clinical studies are essential to confirm its safety and efficacy in human PSCI patients.

## Figures and Tables

**Figure 1 ijms-26-03865-f001:**
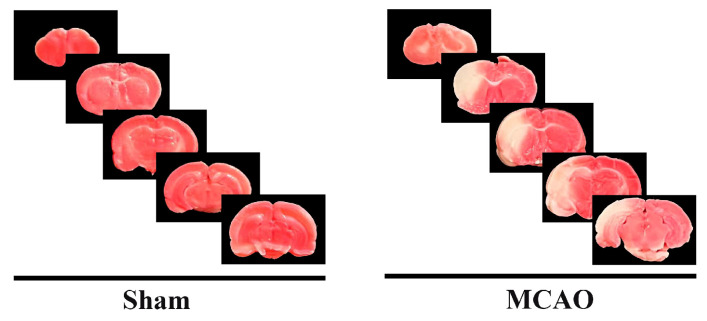
TTC staining was performed to assess cerebral infarction in rats (n = 3/group). The red areas represent viable, normal tissue, while the white areas indicate the regions of cerebral infarction.

**Figure 2 ijms-26-03865-f002:**
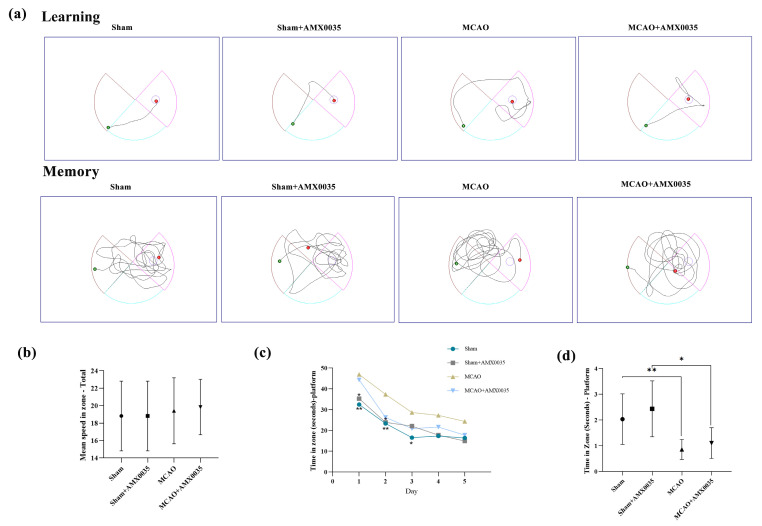
Experimental results from the water maze test in rats from each group. (**a**) Representative motor trajectories during the learning and memory phase of the experiment for each group. (**b**) Average motor velocities of rats on the first day of training in each group. (**c**) Escape latency of rats over 1–5 days of the training period. (**d**) Total time spent on the platform within 1 min after platform removal on day 5. * *p* < 0.05, ** *p* < 0.01.

**Figure 3 ijms-26-03865-f003:**
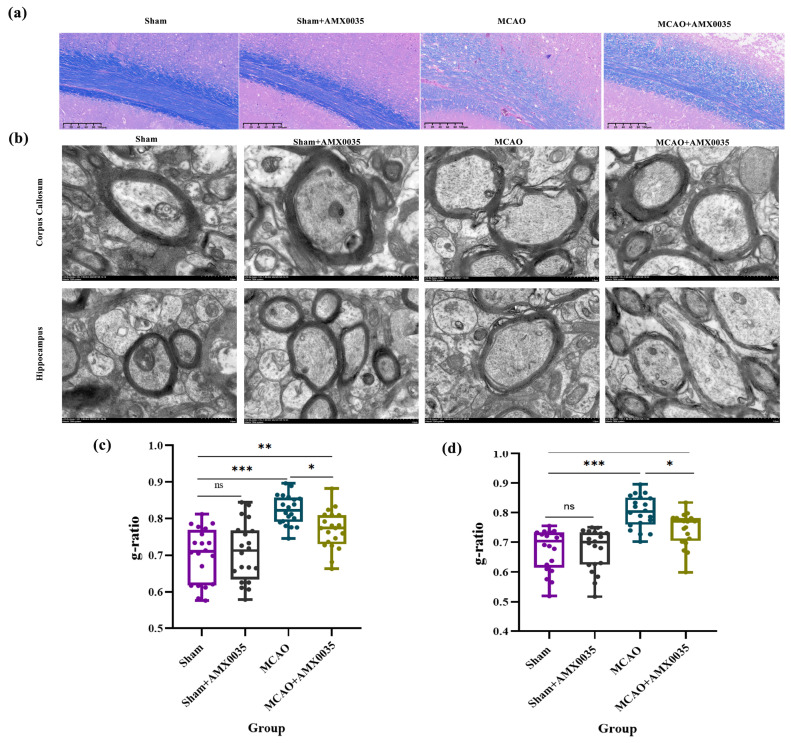
Morphological analysis of the myelin sheath in rats from each group (n = 3/group). (**a**) Myelin structure in the corpus callosum observed using LFB staining. Scale bar = 100 μm. (**b**) Myelin structure in the hippocampus and corpus callosum observed using TEM. Scale bar = 1 μm. (**c**,**d**) Quantitative analysis of the myelin sheath using the g-ratio in the corpus callosum (**c**) and hippocampus (**d**). The g-ratio (axon diameter/total fiber diameter) significantly increased in the MCAO group, indicating thinner myelin relative to the axon size. Treatment with AMX0035 reduced the g-ratio compared to the MCAO group, suggesting the preservation or partial restoration of myelin integrity. Data are presented as the mean ± SD. * *p* < 0.05, ** *p* < 0.01, *** *p* < 0.001, ns = not significant.

**Figure 4 ijms-26-03865-f004:**
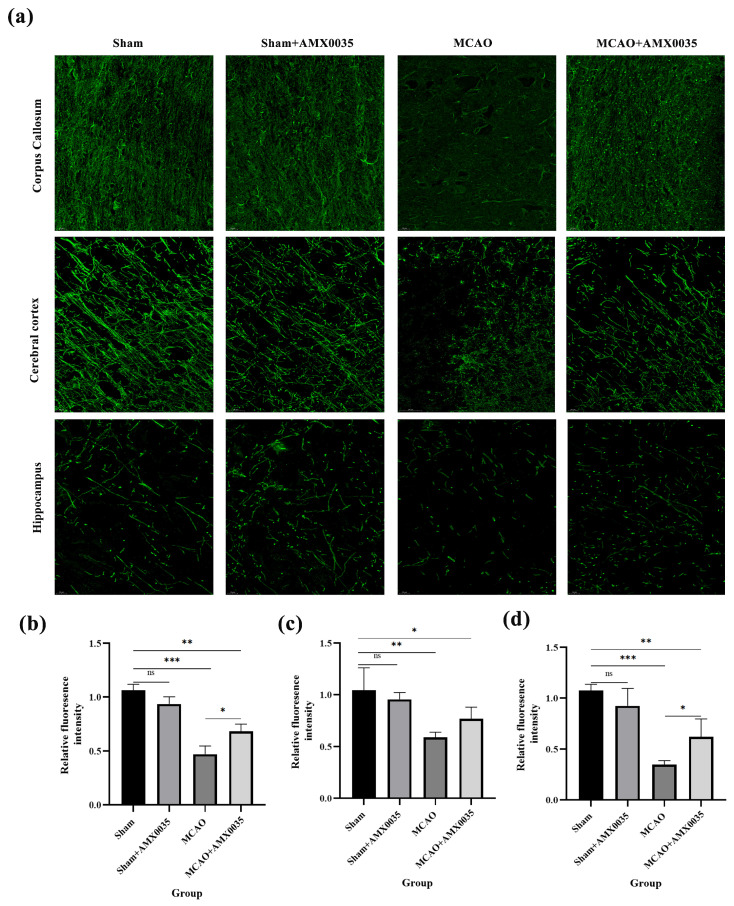
Expression of MBP in the cerebral cortex, corpus callosum, and hippocampus assessed by immunofluorescence (n = 3/group). (**a**) Representative immunofluorescence images showing MBP expression (green) in the cerebral cortex, corpus callosum, and hippocampus of rats from each group. DAPI (blue) was used for nuclear counterstaining. Scale bar = 20 μm. (**b**–**d**) Quantitative analysis of fluorescence intensity in the corpus callosum, cortex, and hippocampus revealed a significant reduction in MBP expression in the MCAO group compared to the sham group, indicating demyelination. Treatment with AMX0035 increased MBP fluorescence intensity in all three regions, suggesting a protective or restorative effect on myelin. Data are presented as the mean ± SD. * *p* < 0.05, ** *p* < 0.01, *** *p* < 0.001, ns = not significant.

**Figure 5 ijms-26-03865-f005:**
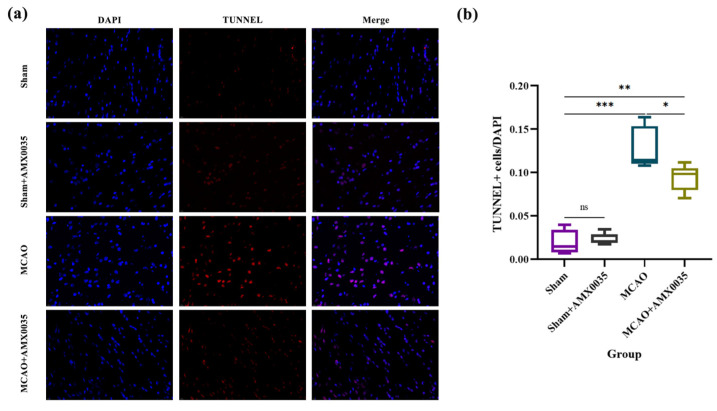
TUNEL staining results in rats from each group (n = 3/group). (**a**) TUNEL staining was used to detect apoptotic cells in the brain tissue sections from each group. Apoptotic nuclei are labeled in red, and all nuclei were counterstained with DAPI (labeled in blue). Scale bar = 10 μm. (**b**) The quantification of TUNEL-positive cells showed a significant increase in apoptosis in the MCAO group compared to the sham group. Treatment with AMX0035 reduced the number of TUNEL-positive cells, indicating an anti-apoptotic effect. Data are presented as the mean ± SD. * *p* < 0.05, ** *p* < 0.01, *** *p* < 0.001, ns = not significant.

**Figure 6 ijms-26-03865-f006:**
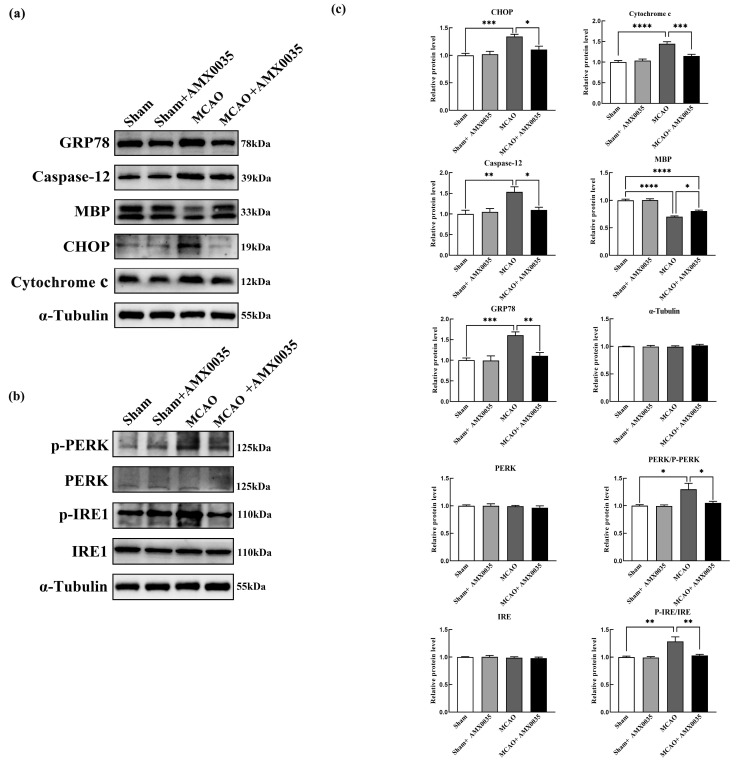
Western blot analysis of ER stress and mitochondrial dysfunction pathway proteins in the corpus callosum of rats from each group (n = 4/group). (**a**,**b**) Representative Western blot images showing the expression levels of key proteins involved in endoplasmic reticulum (ER) stress and mitochondrial dysfunction pathways. (**c**) Quantitative analysis of protein expression levels across groups. Protein expression was normalized to α-tubulin and presented as the mean ±SD. Statistical analysis was performed using one-way ANOVA. * *p* < 0.05, ** *p* < 0.01, *** *p* < 0.001, **** *p* < 0.0001.

**Figure 7 ijms-26-03865-f007:**
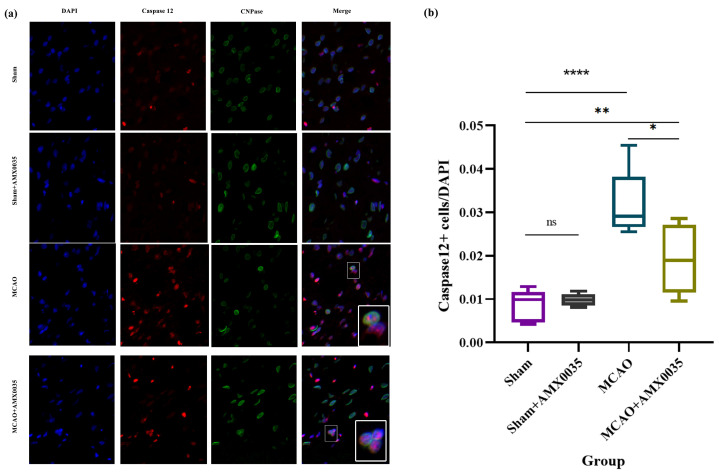
Double-labeling immunofluorescence analysis of caspase 12 and CNPase in rat brain tissue (n = 3/group). (**a**) Representative immunofluorescence images showing the co-localization of caspase 12 (red), a key protein involved in the endoplasmic reticulum (ER) stress pathway, and CNPase (green), a marker of oligodendrocytes, in the brain tissues of rats in each group. Nuclei were counterstained with DAPI (blue). Scale bar = 10 μm. Insets in the lower right corners show high-magnification views of double-positive cells. Scale bar = 5 μm. (**b**) The quantitative analysis of caspase 12 and CNPase double-positive cells revealed a significant increase in ER stress-associated oligodendrocyte apoptosis in the MCAO group, which was markedly reduced following AMX0035 treatment. Data are presented as the mean ± SD. * *p* < 0.05, ** *p* < 0.01, **** *p* < 0.0001, ns = not significant.

**Figure 8 ijms-26-03865-f008:**
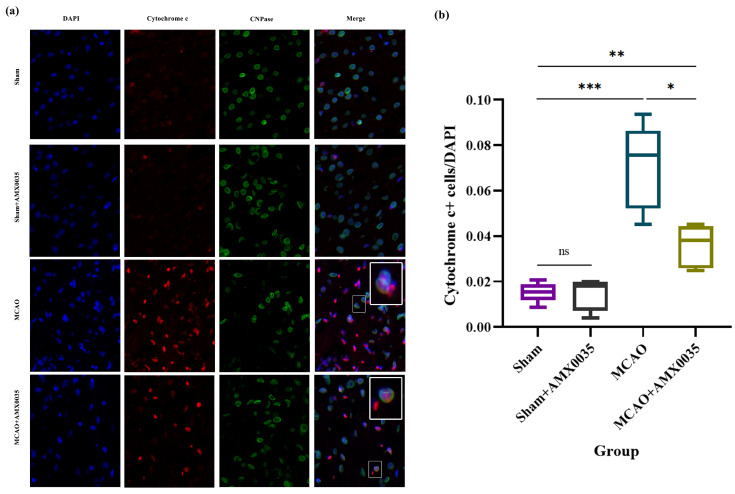
Double-labeling immunofluorescence analysis of cytochrome c and CNPase in rat brain tissue (n = 3/group). (**a**) Representative immunofluorescence images showing the co-localization of cytochrome c (red), a key protein in the mitochondrial dysfunction pathway, and CNPase (green), an oligodendrocyte marker, in the brain tissue of rats in each group. Nuclei were counterstained with DAPI (blue). Scale bar = 10 μm. Insets in the upper right corners show high-magnification views of double-positive cells. Scale bar = 5 μm. (**b**) The quantification of cytochrome c and CNPase double-positive cells across groups. A significant increase in double-positive cells was observed in the MCAO group, while AMX0035 treatment reduced their number. Data are presented as the mean ± SD. * *p* < 0.05, ** *p* < 0.01, *** *p* < 0.001, ns = not significant.

**Figure 9 ijms-26-03865-f009:**
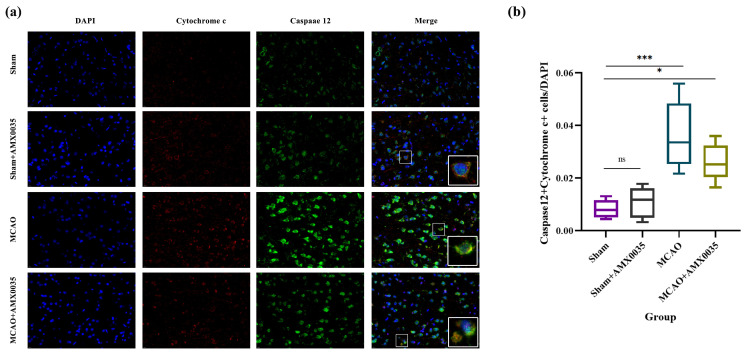
Double-labeling immunofluorescence analysis of caspase 12 and cytochrome c in rat brain tissue (n = 3/group). (**a**) Representative immunofluorescence images showing the co-localization of caspase 12 (green), a key protein in the endoplasmic reticulum (ER) stress pathway, and cytochrome c (red), a mitochondrial dysfunction-related protein, in the brain tissues of rats in each group. Nuclei were counterstained with DAPI (blue). Scale bar = 10 μm. Insets in the lower right corners show high-magnification views of double-positive cells. Scale bar = 5 μm. (**b**) The quantification of caspase 12 and cytochrome c double-positive cells in brain tissue across the groups. The number of double-positive cells significantly increased in the MCAO group and reduced following AMX0035 treatment. Data are presented as the mean ± SD. * *p* < 0.05, *** *p* < 0.001, ns=not significant.

## Data Availability

The data supporting the findings of this study are available from the corresponding author upon reasonable request. These data include all relevant experimental results and analyses, which will be made accessible to researchers interested in verifying or building upon this work.

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
