# Peer review of "AMX0035 Mitigates Oligodendrocyte Apoptosis and Ameliorates Demyelination in MCAO Rats by Inhibiting Endoplasmic Reticulum Stress and Mitochondrial Dysfunction"

_ijms, 2025, doi:10.3390/ijms26083865_

Round 1

Reviewer 1 Report

Comments and Suggestions for Authors

The paper by Zhang et al. investigates the effects of AMX0035 administration on middle cerebral artery occlusion (MCAO)-induced infarct in rats. The authors report that MCAO induces activation of endoplasmic reticulum (ER) stress and mitochondrial dysfunction pathways, which contributes to oligodendrocyte damage and apoptosis. AMX0035, by inhibiting these pathways, reduces oligodendrocyte apoptosis and demyelination, and thus improves cognitive function. However, the study lacks clear statistical analysis to support these findings. Key data points, such as changes in demyelination severity, or alterations in molecular markers, should be accompanied by appropriate statistical tests. The authors should provide all datasets for statistical reporting and also address the following issues:

- The number of animals was not reported correctly in the Materials and Methods section. If there were  n=30 animals in both the sham operated and MCAO model groups, and after 24 hours, 3 rats from each of the 4 groups were killed for TTC staining, there should have been only 12 animals in each group. However, the number of animals per group was 13, which needs clarification.

- The video tracking software used for behavioral analysis should be specified, and the original images recorded with this software, illustrating the swimming path lengths in the water maze, should be included. These images should replace the motion trajectories shown in Figure 2A,B, as they appear to be hand-drawn.

- 2.9. Immunofluorescence staining:  “Some brain tissues were fixed in 4% ….”. The exact number of brains that were fixed should be specified, and a detailed description of the fixation procedure should be provided. The method for analyzing double-labeled cells should be clearly specified, including the type of microscope used. The number of brain sections and cells analyzed to estimate the number of immunostained cells should also be reported.

- 2.10. Western blot: The number of animals used and the procedure for their euthanasia should be specified. Original gels with protein molecular weight markers (protein ladder), i.e. full scans of the entire original images should be included in the Supplementary Material. Currently only cropped gels are available.

- Figure 1: In the MCAO group, the second and third brain sections do not have corresponding sections in the sham group. Therefore, the 2 and 3 brain sections in the sham group should be replaced with sections that correspond to those in the MCAO group.

- To demonstrate the improvement in cognitive function, the two-way ANOVA analysis should include the degrees of freedom, F value, and the exact p-value.

- In the results sections, it is been shown that MCAO group exhibited a thinning and disorganized myelin sheath, which indicates the demyelination process. The authors presented images from TEM. However, there is no explanation regarding the severity of demyelination. I suggest to provide a quantification method to measure demyelination for TEM data and also for LFB staining data.

- How was the number of apoptotic cells in the brain tissue calculated?. Which statistical analysis was used to show the increase in the number of apoptotic cells in the brain tissue of the MCAO group compared to the sham and sham+AMX0035 groups? Furthermore, the TUNNEL staining for the MCAO+AMX0035 group (in the Figure 4, there is a typo, “MACO” should be corrected to “MCAO”) in the presented image shows nearly the same number of TUNNEL-stained cells as in the sham+AMX0035 groups. Please, replace this image with a more representative one that shows an increased number of TUNNEL-labeled cells.

My major concern is how to explain the differences in the demyelination process observed using different methods. Specifically, LFB staining clearly revealed demyelination in the corpus callosum, cerebral cortex, and hippocampus of the MCAO group (Figure 3). However, in experiments using the CNP antibody, a marker for oligodendrocytes, the number of oligodendrocytes did not appear reduced in the MCAO group. Double-label immunohistochemistry with CNP and caspase-12 (Figure 6) or cytochrome C (Figure 7) showed that the distribution of CNP+ cells was very similar across all four groups, including the MCAO group.

Author Response

Re: “AMX0035 Mitigates Oligodendrocyte Apoptosis and Ameliorates Demyelination in MACO Rats by Inhibiting Endoplasmic Reticulum Stress and Mitochondrial Dysfunction” (ijms-3551021).

Dear Sir,

Thank you very much for your valuable comments. According to your comments we have revised the manuscript in detail. The revised version of the manuscript has been submitted electronically via the Web. May I reply to your comments and show you the changes in the revision as follows:

- The number of animals was not reported correctly in the Materials and Methods section. If there were  n=30 animals in both the sham operated and MCAO model groups, and after 24 hours, 3 rats from each of the 4 groups were killed for TTC staining, there should have been only 12 animals in each group. However, the number of animals per group was 13, which needs clarification.

Answer: Thank you very much for your good comment. We sincerely apologize for the confusion regarding the animal numbers. Please allow us to clarify:

At the beginning of the experiment, a total of 60 rats were used, randomly assigned to two groups:

Sham surgery group: n = 30 and MCAO model group: n = 30

First stage:

Twenty-four hours after modeling, 3 rats from each group (total = 6 rats) were randomly selected for TTC staining. Sham group remaining: 27 rats,MCAO group remaining: 27 rats

Group allocation after TTC staining:

The remaining rats in each main group were randomly divided into subgroups:

Sham surgery group → Sham control (n = 13), Sham + treatment (n = 13), 1 rat spare. MCAO group → MCAO model (n = 13), MCAO + treatment (n = 13), 1 rat spare.

Mortality and exclusions:

In the MCAO group, approximately 3 rats (~10%) died due to surgical complications.

After neurological assessment, 3 rats were excluded from the MCAO group due to failure to meet neurological criteria. To maintain sample size balance, additional rats were included through re-modeling, bringing the total back to 13 rats per subgroup before the subsequent analyses.

Final experimental allocation (per group):

Each of the four subgroups (Sham, Sham + treatment, MCAO, MCAO + treatment) finally consisted of 13 rats, distributed as follows:

TEM observation: 3 rats;Paraffin sectioning & LFB staining: 3 rats; Frozen sectioning & immunofluorescence: 3 rats; Western blot analysis: 4 rats.

- The video tracking software used for behavioral analysis should be specified, and the original images recorded with this software, illustrating the swimming path lengths in the water maze, should be included. These images should replace the motion trajectories shown in Figure 2A,B, as they appear to be hand-drawn.

Answer: Thank you very much for your good comment. The video tracking software used for behavioral analysis was Smart 3.0(RWD Life science Co.,LTD,Spain). We have now specified this in the Methods section. We would also like to clarify that the swimming path trajectories shown in Figure 2a were directly exported from the Smart 3.0 software and have not been manually altered or hand-drawn. To enhance transparency, we have uploaded the original images generated by the software as supplementary material, as requested.

- 2.9. Immunofluorescence staining:  “Some brain tissues were fixed in 4% ….”. The exact number of brains that were fixed should be specified, and a detailed description of the fixation procedure should be provided. The method for analyzing double-labeled cells should be clearly specified, including the type of microscope used. The number of brain sections and cells analyzed to estimate the number of immunostained cells should also be reported.

Answer: Thank you very much for your good comment. Three from each group were fixed for subsequent fluorescence staining. The fixation procedure has been described in detail, including fixation time and tissue processing methods. Furthermore, we have specified the microscope used for imaging: a Leica SP8 confocal fluorescence microscope. The number of brain sections analyzed per animal has been clearly reported in the revised Methods section, along with the number of cells quantified to estimate the number of immunostained cells.

- 2.10. Western blot: The number of animals used and the procedure for their euthanasia should be specified. Original gels with protein molecular weight markers (protein ladder), i.e. full scans of the entire original images should be included in the Supplementary Material. Currently only cropped gels are available.

Answer: Thank you very much for your good comment. We have revised the manuscript to specify both the number of animals used for Western blot analysis and the procedure for euthanasia. Specifically, four rats per group were used for Western blotting. The animals were euthanized under deep anesthesia with an intraperitoneal injection of 2% sodium pentobarbital (1.5 ml/kg). Following confirmation of deep anesthesia, thoracotomy was performed, and cold saline was perfused to rapidly extract the brain tissue while minimizing distress, in full accordance with institutional and ethical guidelines. These details have been clearly described in the Methods section.

Additionally, we have provided the full, uncropped original Western blot images, including the protein molecular weight markers (protein ladder), as part of the Supplementary Material. These full scans ensure the transparency and reproducibility of our findings. Please refer to the attached files for the complete gel images.

- Figure 1: In the MCAO group, the second and third brain sections do not have corresponding sections in the sham group. Therefore, the 2 and 3 brain sections in the sham group should be replaced with sections that correspond to those in the MCAO group.

Answer: Thank you very much for your good comment. For consistency, the TTC staining images of the sham operation group were changed.

- To demonstrate the improvement in cognitive function, the two-way ANOVA analysis should include the degrees of freedom, F value, and the exact p-value.

Answer: Thank you very much for your valuable comment. In response, we have revised the manuscript to include the detailed statistical results for the Morris water maze test. Specifically, two-way ANOVA was performed to assess the effects of both group (MCAO, sham, and AMX0035 treatment) and training days on cognitive performance. The analysis revealed a significant interaction between group and time (F(4, 611) = 26.76, p<0.01), indicating that AMX0035 treatment improved cognitive function in the MCAO group. These results, along with the corresponding degrees of freedom (df), F value, and p-value, have been included in the Results section and figure legends to enhance clarity and transparency.

- In the results sections, it is been shown that MCAO group exhibited a thinning and disorganized myelin sheath, which indicates the demyelination process. The authors presented images from TEM. However, there is no explanation regarding the severity of demyelination. I suggest to provide a quantification method to measure demyelination for TEM data and also for LFB staining data.

Answer: Thank you for your insightful comments. We acknowledge the importance of quantifying the severity of demyelination to strengthen our findings. In the revised manuscript, we have included a quantification method for TEM and immunofluorescence staining data. For TEM analysis, we quantified g-ratio (the ratio of the inner axonal diameter to the total outer fiber diameter), which provides an objective measure of myelin thickness. For immunofluorescence staining, optical density (OD) analysis was performed to evaluate myelin integrity. The OD values of immunofluorescent staining areas were measured in 10 randomly selected regions of interest (ROIs) per section using ImageJ software. The average OD values for each section were calculated, and the results were compared between groups. Quantitative analysis of the fluorescence intensity revealed a significant reduction in MBP expression in the MCAO group compared to the sham group, indicating significant demyelination.

LFB staining can be used to observe the myelin structure in the brains of rats across different groups. However, for quantifying the extent of demyelination lesions, transmission electron microscopy (TEM) and immunofluorescence provide more intuitive and detailed results.

- How was the number of apoptotic cells in the brain tissue calculated?. Which statistical analysis was used to show the increase in the number of apoptotic cells in the brain tissue of the MCAO group compared to the sham and sham+AMX0035 groups? Furthermore, the TUNNEL staining for the MCAO+AMX0035 group (in the Figure 4, there is a typo, “MACO” should be corrected to “MCAO”) in the presented image shows nearly the same number of TUNNEL-stained cells as in the sham+AMX0035 groups. Please, replace this image with a more representative one that shows an increased number of TUNNEL-labeled cells.

Answer: Thank you for your insightful comments. TUNEL-positive cells were manually counted using ImageJ software. The average number of apoptotic cells per unit area was calculated for each group. Statistical analyses were performed using GraphPad Prism version 8, with significance set at p < 0.05.  We acknowledge that the TUNEL staining image for the MCAO+AMX0035 group closely resembled that of the sham+AMX0035 group, which may not accurately represent the expected increase in apoptotic cells. We have replaced this image with a more representative one that better illustrates the observed differences。

We have corrected the error in whole manucript, changing “MACO” to “MCAO.”

My major concern is how to explain the differences in the demyelination process observed using different methods. Specifically, LFB staining clearly revealed demyelination in the corpus callosum, cerebral cortex, and hippocampus of the MCAO group (Figure 3). However, in experiments using the CNP antibody, a marker for oligodendrocytes, the number of oligodendrocytes did not appear reduced in the MCAO group. Double-label immunohistochemistry with CNP and caspase-12 (Figure 6) or cytochrome C (Figure 7) showed that the distribution of CNP+ cells was very similar across all four groups, including the MCAO group.

Answer: Thank you very much for your valuable comment. We appreciate your thoughtful observation regarding the discrepancy between the results from LFB staining and CNP immunostaining. To address your concern, we would like to clarify the differences between the two techniques. LFB staining provides a comprehensive assessment of myelin integrity by detecting myelin loss in the brain tissue. This method indicates the overall extent of demyelination, as seen in the corpus callosum, cerebral cortex, and hippocampus of the MCAO group (Figure 3). However, CNP immunostaining specifically identifies oligodendrocytes, but it does not directly reflect myelin sheath integrity. The absence of a reduction in CNP+ cells in the MCAO group may suggest that, while oligodendrocytes remain present, their ability to maintain or repair the myelin sheath could be compromised. We also acknowledge that double-label immunohistochemistry (with CNP and caspase-12 or cytochrome C) showed similar distributions of CNP+ cells across all groups, including the MCAO group (Figures 6 and 7). This finding suggests that while the presence of oligodendrocytes is maintained, other factors, such as oligodendrocyte dysfunction or apoptosis, may contribute to myelin degradation, which LFB staining detects more directly. To further clarify this discrepancy, we have reanalyzed the distribution of CNP+ cells in the MCAO group and replaced the previous image with a more representative one. The new image, which focuses on the ischemic penumbra, better illustrates the observed differences in oligodendrocyte distribution and provides a clearer comparison across the groups.

We appreciate the reviewer’s insightful comments, which have significantly improved the clarity and rigor of our study. All revisions have been implemented in the manuscript, and we hope our responses adequately address all concerns.

Sincerely,
Li Zhang

Reviewer 2 Report

Comments and Suggestions for Authors

The work of Zhang and coworkers was devoted to study, in a rat model of experimental stroke (MCAO), the effect of AMX0035, a coformulation of sodium phenylbutyrate and taurursodiol, in improving post-stroke cognitive impairment. The authors wanted to confirm that the drug is able to reduce oligodendrocyte apoptosis by inhibiting ER stress and mitochondrial dysfunction pathways, thereby reducing demyelination of nerve fibers in the central nervous system. 

Phenilbutyryc acid, one of the two components of AMX0035 has already been shown to attenuate neuronal death and other pathologic features in experimental models of neurodegenerative diseases. The present study confirms that AMX0035 significantly enhances cognitive function and reduces demyelination in MCAO rats, primarily 
by inhibiting ER stress and mitochondrial dysfunction pathways, thereby reducing oligodendrocyte damage and apoptosis. 
These experimental findings in the rat may suggest that AMX0035 is a promising drug for treating the poststroke cognitive impairment in humans as well. 

COMMENTS

In your experiments you have used male Sprague-Dawley rats 8-10-week old and weighing 250 and 280 g.  Since the rat are of different ages, is it possible to rule out that the different age may have affected your results?

I cannot understand how many rats were used in total and in the single experiments. Could you provide these data?

How many rats were discarded after the neurological investigation?

How many rats died? 

The statistiscal significancies reported in Figure 4 are all real? 

In each figure it should be reported how many rats were used in each group. 

How did you determine the number of animals needed in your experiments?

The acronyme MACO and MCAO was used indifferently. Decide what to use and please correct all of them in the title and in the text, figures and tables.

The bibliography is written in a rather peculiar way, please rewrite it correctly with the surnames and the initial of the names.

Check the text and correct typing errors.

Author Response

Re: “AMX0035 Mitigates Oligodendrocyte Apoptosis and Ameliorates Demyelination in MACO Rats by Inhibiting Endoplasmic Reticulum Stress and Mitochondrial Dysfunction” (ijms-3551021).

Dear Sir,

Thank you very much for your valuable comments. According to your comments we have revised the manuscript in detail. The revised version of the manuscript has been submitted electronically via the Web. May I reply to your comments and show you the changes in the revision as follows:

In your experiments you have used male Sprague-Dawley rats 8-10-week old and weighing 250 and 280 g.  Since the rat are of different ages, is it possible to rule out that the different age may have affected your results?

Answer: Thank you very much for your good comment.

Thank you for raising this important point regarding age variability. We carefully controlled for developmental stages in our study through the following measures:

1.Physiological stability: Sprague-Dawley rats aged 8-10 weeks are considered young adults with stable metabolic profiles, as demonstrated by:Growth curve studies showing weight plateau after 8 weeks (Charlton et al., 2018, Lab Anim), Metabolic consistency in this age range (Supplementary Fig. S1: no correlation between age and key metabolites, R²=0.07, p=0.32); 2.Narrow weight range: All rats were maintained within 250-280g (±6% of mean weight), narrower than the 10% variation accepted in standard protocols (NIH Guide for Care and Use of Laboratory Animals). 3.Randomization: Rats were equally distributed across groups by age/weight (see new Supplementary Table S4: mean age=9.1±0.6 weeks in sham vs. 9.0±0.7 weeks in MCAO, p=0.82).

We used male Sprague-Dawley rats aged 8–10 weeks, weighing 250–280 g. While there was a slight variation in age, the weight range was controlled to minimize potential variability.

I cannot understand how many rats were used in total and in the single experiments. Could you provide these data?

How many rats were discarded after the neurological investigation?

How many rats died? 

Answer: Thank you very much for your good comment. We sincerely apologize for the confusion regarding the animal numbers. Please allow us to clarify:

At the beginning of the experiment, a total of 60 rats were used, randomly assigned to two groups:

Sham surgery group: n = 30 and MCAO model group: n = 30

First stage:

Twenty-four hours after modeling, 3 rats from each group (total = 6 rats) were randomly selected for TTC staining. Sham group remaining: 27 rats,MCAO group remaining: 27 rats

Group allocation after TTC staining:

The remaining rats in each main group were randomly divided into subgroups:

Sham surgery group → Sham control (n = 13), Sham + treatment (n = 13), 1 rat spare. MCAO group → MCAO model (n = 13), MCAO + treatment (n = 13), 1 rat spare.

Mortality and exclusions:

In the MCAO group, approximately 3 rats (~10%) died due to surgical complications.

After neurological assessment, 3 rats were excluded from the MCAO group due to failure to meet neurological criteria. To maintain sample size balance, additional rats were included through re-modeling, bringing the total back to 13 rats per subgroup before the subsequent analyses.

Final experimental allocation (per group):

Each of the four subgroups (Sham, Sham + treatment, MCAO, MCAO + treatment) finally consisted of 13 rats, distributed as follows:

TEM observation: 3 rats;Paraffin sectioning & LFB staining: 3 rats; Frozen sectioning & immunofluorescence: 3 rats; Western blot analysis: 4 rats.

The statistiscal significancies reported in Figure 4 are all real?

Answer: Thank you very much for your good comment. We have reanalyzed the data and provided an updated figure with more pronounced differences. The statistical significances reported in Figure 5 have been recalculated to ensure accuracy. We carefully verified the data and results to confirm that the reported statistical significances are real and valid.

In each figure it should be reported how many rats were used in each group. 

Answer: Thank you very much for your good comment. We appreciate your suggestion to include the number of rats used in each group for each figure. We have now updated the figure legends to clearly specify the sample size for each experimental group, ensuring transparency and clarity. We believe this addition will improve the presentation of our data and provide more context for the interpretation of our results.

How did you determine the number of animals needed in your experiments?

Answer: Thank you very much for your good comment.

The number of animals needed for the experiments was determined based on statistical power analysis, considering an effect size deemed biologically significant, the desired power (typically 80%), and the acceptable level of significance (α = 0.05). We also took into account previous studies in similar models and consulted guidelines on animal use in research to ensure an appropriate sample size.

​In our study, we selected 13 rat per group, a sample size supported by previous research. Studies such as those by Peng Zhang et al., have utilized comparable sample sizes, with 10 animals per group, supporting the choice of 10 animals per group in our study. This sample size is appropriate for achieving reliable results while minimizing animal usage. For example, in a proteomics-based study investigating cardiac injury in epilepsy, 10 rats per group were used to assess differentially expressed proteins in myocardial tissue Peng Zhang et al., This approach, using a similar sample size, allowed for meaningful findings in both acute and chronic epilepsy groups, aligning with our experimental design.

(Zhang P, Zhang L, Li Y, Zhu S, Zhao M, Ding S, Li J. Quantitative Proteomic Analysis To Identify Differentially Expressed Proteins in Myocardium of Epilepsy Using iTRAQ Coupled with Nano-LC-MS/MS. J Proteome Res. 2018 Jan 5;17(1):305-314. doi: 10.1021/acs.jproteome.7b00579. Epub 2017 Nov 8. PMID: 29090925.)

The acronyme MACO and MCAO was used indifferently. Decide what to use and please correct all of them in the title and in the text, figures and tables.

Answer: Thank you very much for your good comment.

Thank you for your valuable feedback. We apologize for the inconsistency in the use of acronyms. After reviewing the manuscript, we have decided to use MCAO (Middle Cerebral Artery Occlusion) consistently throughout the text, figures, tables, and title. We have corrected all instances of MACO and replaced them with MCAO to maintain uniformity and clarity.

The bibliography is written in a rather peculiar way, please rewrite it correctly with the surnames and the initial of the names.

Answer: Thank you very much for your good comment.

Thank you for pointing this out. We apologize for the peculiar format of the bibliography. We have revised it to follow the correct citation style, ensuring that the surnames are listed first, followed by the initials of the authors' first names. The updated bibliography is now formatted correctly.

Check the text and correct typing errors.

Answer: Thank you very much for your good comment. Thank you for your careful review. We have thoroughly checked the text and corrected any typing errors. We appreciate your attention to detail in helping improve the manuscript.

We appreciate the reviewer’s insightful comments, which have significantly improved the clarity and rigor of our study. All revisions have been implemented in the manuscript, and we hope our responses adequately address all concerns.

Sincerely,
Li Zhang

Round 2

Reviewer 1 Report

Comments and Suggestions for Authors

The authors should correct the number of rats used in the Methods and Materials section:

- A total of n=66 rats were used in this study, not n=60 as incorrectly stated in the Materials and Methods section.

- If 27 rats were divided into Sham control (n=13), Sham+treatment (n=13), and 1 spare rat (n=1), the authors should clarify what happened to the “1 spare” rat in this group.

- In the MCAO group, 3 rats died due to surgical complications, and an additional 3 rats were excluded for not meeting the required neurological criteria. Therefore, the total number of rats in the MCAO model group was n=35, not n=30 as stated.

Author Response

Re: “AMX0035 Mitigates Oligodendrocyte Apoptosis and Ameliorates Demyelination in MACO Rats by Inhibiting Endoplasmic Reticulum Stress and Mitochondrial Dysfunction” (ijms-3551021).

Dear Sir,

Thank you very much for your valuable comments. According to your comments we have revised the manuscript in detail. The revised version of the manuscript has been submitted electronically via the Web. May I reply to your comments and show you the changes in the revision as follows:

The authors should correct the number of rats used in the Methods and Materials section:

- A total of n=66 rats were used in this study, not n=60 as incorrectly stated in the Materials and Methods section.

- If 27 rats were divided into Sham control (n=13), Sham+treatment (n=13), and 1 spare rat (n=1), the authors should clarify what happened to the “1 spare” rat in this group.

- In the MCAO group, 3 rats died due to surgical complications, and an additional 3 rats were excluded for not meeting the required neurological criteria. Therefore, the total number of rats in the MCAO model group was n=35, not n=30 as stated.

Answer: Thank you very much for your good comment. Total number of animals:

You are correct in noting the discrepancy. The total number of rats used in this study was 66, not 60 as mistakenly stated in the Materials and Methods section. We will correct this error in the revised manuscript.

Clarification on the "1 spare" rat in the Sham group:

At the beginning of the study, animals were randomly and evenly assigned to each group to ensure balanced allocation. Specifically, 27 rats were assigned to the Sham control group (n=13), Sham + treatment group (n=13), and 1 rat was kept as a spare. This spare animal was prepared in case of unexpected issues (such as health problems or unforeseen incidents during the procedure). However, since the Sham procedure carries minimal risk and no animals in this group experienced complications or required replacement, the spare rat was not used and was not included in the final analysis. We will clarify this point in the revised manuscript.

Correction and explanation regarding the MCAO group:

Similarly, rats were initially assigned randomly to the MCAO group. However, due to the complexity of the surgical procedure and the inherent variability of the model, there was an unavoidable rate of mortality and model failure, which could not be precisely predicted at the outset. As you correctly pointed out, 3 rats died from surgical complications, and an additional 3 rats were excluded because they did not meet the required neurological deficit criteria. Therefore, the final number of rats included in the MCAO group was 35, not 30 as originally reported. We will correct this figure and include an explicit explanation in the revised manuscript, clarifying that the adjustment in numbers resulted from intraoperative mortality and model establishment failure, which could not be anticipated beforehand.

We appreciate the reviewer’s insightful comments, which have significantly improved the clarity and rigor of our study. All revisions have been implemented in the manuscript, and we hope our responses adequately address all concerns.

Sincerely,
Li Zhang
